# CSFCube – A Test Collection of Computer Science Research Articles for Faceted Query by Example

**Sheshera Mysore[1]**     **Tim O'Gorman[2]***     **Andrew McCallum[1]**     **Hamed Zamani[1]**

{smysore, mccallum, zamani}@cs.umass.edu

[1]University of Massachusetts, Amherst, MA, USA
[2]Thorn, CA, USA

## Abstract

Query by Example is a well-known information retrieval task in which a document is chosen by the user as the search query and the goal is to retrieve relevant documents from a large collection. However, a document often covers multiple aspects of a topic. To address this scenario we introduce the task of *faceted Query by Example* in which users can also specify a finer grained aspect in addition to the input query document. We focus on the application of this task in scientific literature search. We envision models which are able to retrieve scientific papers analogous to a query scientific paper along specifically chosen rhetorical structure elements as one solution to this problem. In this work, the rhetorical structure elements, which we refer to as *facets*, indicate objectives, methods, or results of a scientific paper. We introduce and describe an expert annotated test collection to evaluate models trained to perform this task. Our test collection consists of a diverse set of 50 query documents in English, drawn from computational linguistics and machine learning venues. We carefully follow the annotation guideline used by TREC for depth-k pooling (k = 100 or 250) and the resulting data collection consists of graded relevance scores with high annotation agreement. State of the art models evaluated on our dataset show a significant gap to be closed in further work. Our dataset may be accessed here: `https://github.com/iesl/CSFCube`

## 1   Introduction

The dominant paradigm of information retrieval is to treat queries and documents as different kinds of objects, e.g., in keyword search. This paradigm, however, does not lend itself to exploratory search tasks. On the other hand, paradigms of search such Query by Example (QBE) which treat queries and documents as similar kinds of objects have been considered more suited to exploratory search tasks [41, 17, 45]. QBE has also been used more recently in information extraction for NLP [61, 58], and seems poised to leverage recent advances in representation learning and NLP for search tasks where keyword search proves insufficient [6, 70].

In QBE search tasks, each query is a document that often covers multiple aspects of a topic leading to a document-only query under-specifying how retrievals should me made. Here, we introduce the task of faceted QBE, where users can specify an information need by providing an input document and a facet example, with the goal to retrieve documents that are similar to the input document from the perspective of the given facet. This paper focuses on the case of faceted QBE applied to scientific articles. Figure 1 illustrates how multi-aspect similarities may arise in scientific articles, where candidate documents could be similar to the query along the general problem being addressed or the method used in a paper.

---

*Work done while at the University of Massachusetts, Amherst.

35th Conference on Neural Information Processing Systems (NeurIPS 2021) Track on Datasets and Benchmarks.

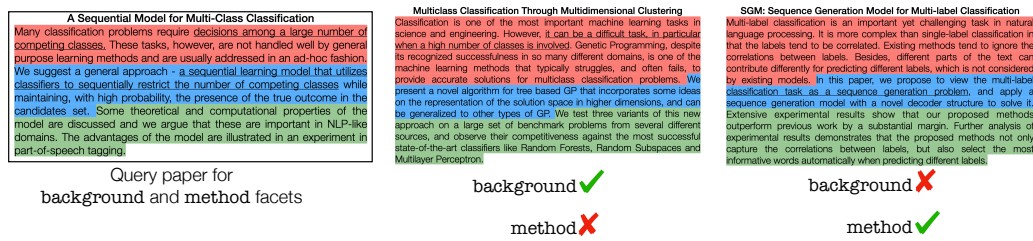

Figure 1: Examples of candidate papers similar along different facets for the same query paper, underlined text indicates similar aspects: A paper discussing a classification problem with large number of classes is similar along `background` with a paper discussing the same problem, but with a different paper discussing a sequential model for its `method` facet.

The promise of literature search to navigate vast collections of research documents offers exciting prospects and the potential to accelerate the process of scientific discovery [24]. This optimism has been expressed in a range of work in biomedicine [67], materials science [27], geography [42], biomimetic design [40], and machine learning [66]. The ever-growing research literature has also lead to search and recommendation tools being important in the workflows of individual researchers across a range of disciplines [43, 55]. However, few literature search and recommendation tools address the problem that documents often contain multiple fine grained aspects one might want to search with [53]. Further, prior work has also demonstrated that researchers often desire finer-grained control in literature search and exploration tools [18, 51, 28]. Context-dependent faceted search tools have the potential to fill this gap. Finally, while we focus on literature search, faceted QBE also has the potential to serve in other applications such as e-commerce search and product design [29].

The problem at the heart of faceted QBE is that of fine-grained document similarity. This is of broader relevance to numerous other research problems. This work therefore promises to be of value in research problems such as that of multi-aspect reviewer assignments for papers [35], tracing the adoptions of ideas from the research literature [11], or that of making causal inferences with matched scientific texts [57, Sec: Applications and Simulations]. Appendix E elaborates on applications.

Despite this wide applicability of the problem, and a range of proposed approaches, evaluation of these document retrieval/similarity methods remains a problem – with prior work relying on weak sources of gold evaluation data or on expensive human evaluations. A broad set of approaches evaluate systems against citations or combine citation information with other incidental information such as section headers as a means to determine citation intents acting as proxies for facets [54]. While these sources may represent reasonable noisy signals [33, Sec 4] for model training, the noise in such approaches limits their value as a method of system evaluation (Appenxix D presents an analysis of citations as evaluation data). Other work has relied on extrinsic evaluations in a downstream ideation task [13] or based on feedback of users interacting with a recommendation system [12, 1]. However, such approaches require expensive evaluation protocols unsuitable for model development and system comparisons. Therefore, we believe our dataset will fill an important gap by presenting a pragmatic alternative to these extremes.

Our publicly accessible test collection consists of 50 diverse research paper abstracts in English paired with a facet as queries (§3). The query facets are chosen from one among three facets from $\mathcal{F} = \{\texttt{background/objective}, \texttt{method}, \texttt{result}\}$, representing the three dominant aspects of methodological scientific research. Query abstracts are selected from the domains of machine learning and natural language processing. Candidate pools of size 250 (8 queries) or 100 (42 queries) are constructed from about $800,000$ computer science papers in the S2ORC Corpus [46] using a diverse set of retrieval methods. Four graduate-level computer science annotators with research experience rated each candidate with respect to the query abstract and facet, with high agreement. Finally, we also present a range of baseline experiments and analysis (§5) indicating the phenomena that state of the art models fail to capture, presenting significant room for improvement.

## 2    Task Formulations

We frame our task as one of retrieving scientific papers given a query paper and additional information indicating the query facet. In this work we operate at the level of abstracts rather than the body text of papers under the understanding that salient information about a paper is contained in the abstract, especially for the domains considered in this paper [30]. Further, important applications in large deployed systems for paper recommendation and reviewer assignment operate at the level of paper abstracts indicating them to be an already useful choice [60, 52]. Our decision is also motivated by the difficulty of annotating large corpora at the body-text level, and mirrors a common choice in numerous prior work [67, 10]. Note however, that our dataset is structured to leverage future full-text approaches (See §3). Finally, §4 presents a small-scale analysis of full-text vs abstract annotation consistency. In what follows, we also assume access to the title of the papers, although we drop it from the below discussion for brevity.

We denote a query document with $Q$, a candidate document $C \in \mathcal{C}$. Every $Q$ or $C$, consist of $N$ sentences $\langle S_1, S_2, \ldots S_N \rangle$. We denote *facets*, $f$ from an inventory of labels $\mathcal{F}$, indicating the rhetorical elements of the document. We denote a ranked list of the candidates for $Q$ and query facet $f_q$ with $\langle r, C \rangle \in R_{Qf}$ where elements denotes document $C$ at rank position $r$. Now, we formulate two tasks that our test collection will effectively evaluate:

**Definition 1.** *Retrieval based on pre-defined facets: Given query and candidate documents – $Q$ and $\mathcal{C}$, with sentences in both annotated with facet labels: $\langle (f_1, S_1), (f_2, S_2), \ldots (f_N, S_N) \rangle$ and a query facet $f_q$ a system must output the ranking $R_{Qf}$.*

**Definition 2.** *Retrieval by sentences: Given query and candidate documents – $Q$ and $\mathcal{C}$, and a subset of sentences $\mathcal{S}_Q \subseteq Q$ based on which to retrieve documents a system must output the ranking $R_Q$.*

Def 1 corresponds most closely to faceted search as described by Kong [37] and closest to work by Chan et al. [13]. While Def 2 represents a more general formulation not relying on pre-specified facet labels. Here, the sentences $\mathcal{S}_Q$ can be viewed as exemplar facet sentences based on which results must be retrieved even while lacking any explicit facet specification.

One may view both Definition 1 and 2 as instances of the QBE paradigm of retrieval [45]. One, at the level of documents, using the document $Q$ as a query as in El-Arini and Guestrin [19] and Zhu and Wu [70], and a second at the level of sentences denoting a facet of the paper. Broadly, we believe QBE to be well suited to the problem of faceted literature search given the difficulty of being able to specify in keyword searches precisely the search intent and given that the meaning of sentences denoting a facet are often dependent on the broader context of the abstract. Further, we expect Definition 2 to have specific other advantages: users often tend to have different understanding of facets than those defined by designers of the ranking system [64] – in our case we expect that different sub-areas/areas of the literature will exemplify different kinds of facets making it hard to pre-specify facets in a system. Further, users often wish to explore the literature at different levels of granularity than that possible with pre-defined facets [28, page 14], we expect QBE will allow users greater control to select parts of an abstract expressing a facet at different levels of granularity based on how they would like papers retrieved. Importantly however, note that while our dataset selects a specific set of facets for ease of annotation it facilitates evaluation and consequent model development of both task setups.

## 3    Dataset Description

In the construction of this test collection we relied on the Semantic Scholar Open Research Corpus (S2ORC) [46] which provides a corpus of $81.1M$ English language research papers alongside a range of automatically generated metadata including citation network information. We choose to work with about $800,000$ computer science papers in S2ORC sourced from arXiv.[2] These papers were selected to ensure that the full-body text of the papers was available, in addition to the abstract and title, to facilitate potential future research.[3] Our queries were selected from domains of machine learning and NLP so that annotators would be familiar with the domain in question.

---

[2] `https://arxiv.org/`

[3] `datasheet.md` in the dataset release documents detailed filtering steps used to obtain the 800,000 documents. Our release also includes these 800,000 documents.

**Facets:** In this work a facet for a research paper corresponds to the dominant steps involved in carrying out scientific research – the identification of a research problem/question (`background/objective`), formation and testing of the hypothesis (`method`), and formation of conclusions (`results`). These facets are broadly defined as:

`background/objective`: Most often sets up the motivation for the work, states how it relates to prior work and states the problem or research question being asked. Henceforth, we refer to these as `background` facets.

`method`: Describes the method being proposed or used in the paper. The method could be described at a very high level or it might be specified at a very fine-grained level depending on the type of paper. Note that our definition of methods is broad and will include methods of analysis of a phenomena, a model, data, or procedural descriptions of the experiments carried out. The specific interpretation of method also depends on the type of paper (§3).

`result`: This may be a detailed statement of the findings of analysis, a statement of results or a concluding set of statements based on the type of paper.

Our corpus is labelled with facets predicted using the model of Cohan et al. [15] into the set of labels: {`background`, `objective`, `method`, `result`, `other`}. Incorrect facet labels for the query abstracts are manually corrected. Prior to relevance annotation, `objective` and `background` are merged into one facet called `background` as they were too similar to be distinguished for the purpose of document similarity. The `other` facet is not considered for annotation. These sentence facets were then provided as additional guidance during annotation, with query facet sentences being bolded to encourage attention to those parts of the document.

**Query Abstract Selection:** We annotated a total of 50 query abstract-facet pairs from the ACL Anthology.[4] Of the 50, we annotated 16 abstracts with two different facets each (total of 32 query abstract-facet pairs), in order to allow closer analysis of the differences in retrieval performance for the same query abstract while varying the query facet (Figure 3). The remaining 18 abstracts were annotated for a single query facet each. In total, our dataset contains 16 `background` queries, 17 `method` queries, and 17 `result` queries; further statistics are provided in Table 1. Queries were selected to ensure coverage over a range of years (1995-2019) and to ensure a somewhat even distribution across query paper types, as coarsely divided into "resource/evaluation papers", "data-driven approach papers" or "theoretical papers". This was ensured through randomly sampling a set of 100 articles over the time period and were manually filtered to ensure that each query corresponded to specific and non-trivial representations of multiple facets. We include the query data distributions and the procedure for query selection in the annotator guidelines in our dataset release.[5]

**Candidate Pooling:** Candidates per query are drawn from a corpus of about 800,000 computer science papers in the S2ORC corpus using the following pool of methods: TF-IDF, averaged `word2vec` embeddings, and TF-IDF weighted `word2vec` based similarities run on titles, and abstracts giving us a set total of 6 methods. Further, a state-of-the-art BERT model, SPECTER [16], trained for scientific paper representation using citation network signals was also part of the set of methods used to generate our pool. Finally, papers cited in the query paper are also added to the pool, given their likely relevance due to authors self selections. This set of methods represents a diverse range of similarities with each of the methods retrieving largely different candidate abstracts: The top-25 papers across retrieval methods contained between 1-4 papers in common. For a set of 8 abstract-facet queries, we annotate pools of size 250. These formed an initial exploratory set of annotations, and the remaining 42 queries were annotated with pools roughly of size 100. For queries with pools of size 250 we draw the top 33 papers from each retrieval method, similarly for queries with pools of 100 we draw the top 13 papers. The order in which we draw from the group of methods is randomized for every query and in the case of a candidate already present in our candidate pool we draw from further down the ranked list of a method. Finally, while our task is framed as facet dependent, we use non-faceted methods in the construction of our pool due to the lack of well-established faceted retrieval models. This choice also allowed us to ensure an identical pool of candidates for being annotated with respect to different facets – providing for a richer evaluation setup. Statistics of the dataset are provided in Table 1.

---

[4]`https://www.aclweb.org/anthology/`

[5]Annotator guideline and query metadata files in our dataset release: `ann_guidelines.pdf` and `queries-release.csv`

Table 1: Statistics for the test collection.

| Statistic | | All |
|---|---|---|
| Query abstract-facet pairs | - | 50 |
| Unique query abstracts | - | 34 |
| Mean candidate pool size | - | 124.9 |
| Query-candidate pairs | - | 6244 |
| | min | 12 |
| Candidates rated +1 per query | max | 87 |
| | avg | 36.9 |
| | min | 1 |
| Candidates rated +2/+3 per query | max | 35 |
| | avg | 9.8 |

Table 2: Spearman's $\rho$, Krippendorff's $\alpha$, Cohen's $\kappa$, and % agreement measures for relevances before and after the adjudication stage of annotation. Given the ranking nature of the task, Spearman's $\rho$ presents the most apt measure of agreement.

| facet | pre-adjudication | | | |
| | $\rho$ | $\alpha$ | $\kappa$ | % |
|---|---|---|---|---|
| background | 0.45 | 0.43 | 0.28 | 57.07 |
| method | 0.31 | 0.26 | 0.20 | 69.60 |
| result | 0.42 | 0.35 | 0.26 | 67.46 |
| | post-adjudication | | | |
| background | 0.73 | 0.72 | 0.62 | 77.68 |
| method | 0.63 | 0.61 | 0.54 | 84.47 |
| result | 0.70 | 0.67 | 0.59 | 83.53 |

## 4  Dataset Annotation

**Relevance Ratings:**  We choose to rate candidate documents on a graded scale from 0-3 with our definitions for the scales depending on the facet. Broadly, we train annotators to rate structural/relational similarities between the candidate and query higher (3-2 ratings) than attribute/feature based similarities (1-0 ratings). This draws on motivations from a range of literature highlighting the importance of structural similarities between ideas to creative activities like scientific research – a focus of this work [47, 23, 13, 44]. We illustrate the definitions with the `method` facet here:

`Near Identical/+3`: +3 implies methods described share a similar over-arching mechanistic similarity, further the methods must also be similar in terms of the details of the objects being manipulated.

`Similar/+2`: +2 implies that the methods are mechanistically similar and the details are only comparable between the query and candidate.

`Related/+1`: +1 is meant to encompass a wide range in being similar and can be hard to list at length. Common cases include: 1. Details of the two methods are similar but there is only high level mechanistic similarity. 2. Small or not-so-important mechanistic parts of the methods in two papers are similar. 3. Where query and candidate abstracts may vary in the level of granularity in which they describe a method and a high level similarity is the only one you can establish by reading the abstract.

Our relevance grades also include an `Unrelated/0` grade for documents deemed unrelated. We encourage readers to examine `ann_guidelines.pdf` in our dataset release which details these further alongside examples for every case.

**Annotation Procedure:**  Our annotation was carried out by four graduate-level computer science annotators (the lead authors, SM, TO, and 2 hired annotators) with experience reading research papers in the selected domains.[6] The annotation guidelines were developed over 4 iterations of repeated annotation and refinement of the guidelines. The hired annotators were trained prior to annotation and demonstrated Spearman Correlation based agreements of $0.5 - 0.7$ with an adjudicated training set of examples. Annotators were paid an hourly wage of USD 22.5 for a period of 3 weeks. All query-candidate pairs were annotated by two independent annotators and a third adjudicator resolved the cases of difference between the first two annotators. All annotations were carried out in Label Studio[7] and annotators were only shown paper titles and abstracts at all stages of annotation, hiding all other metadata including authors, publication venues, and years.

**Relevance Analysis:**  Given our two stage annotation process of gathering double annotation and an adjudication stage, we report agreement metrics for both the stages. Table 2 presents these agreements. Our pre-adjudication metrics are those between the two annotators involved in the annotation. For the post-adjudication metrics, we report the mean metrics between the adjudicated ratings and each of the two initial annotators ratings. We report Spearman rank-correlation coefficients, $\rho$, between

---

[6]Further details about annotators are included in the `datasheet.md` in the dataset release.
[7]`https://labelstud.io/`

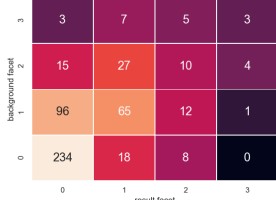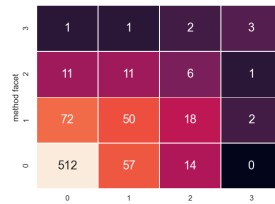

Figure 2: Number of candidates labeled with a particular facet relevance (scale of 0-3) for the documents labeled with two facets. Note that while some between facets correlation exists (along the diagonals), many candidates are relevant only to one facet. Also note that this differs by facet pair.

annotators (pre-adjudication) and between annotators and adjudicated rankings (post-adjudication) produced by ratings. We believe $\rho$ to be the most apt measure of agreement given our ranking task, where we are most interested in establishing a relative similarity between papers. Our reporting also follows work in rating sentence similarities which reports correlations between annotators as a measure of agreement [3]. Additionally, we report Krippendorff's $\alpha$ with an ordinal distance function to measure agreement in absolute terms while taking into account an ordinal relevance scale, Cohens $\kappa$ to measure agreement while not taking into account the ordinal nature of relevance levels i.e. treating ratings as categorical labels, and simple percent agreements as an illustration of the fraction of data which needed adjudication. All of the metrics in Table 2 represent median values across the per-query metrics. It was also permissible for the adjudicator to entirely over-rule both annotators ratings for a candidate. Across all facets, we saw this happen very rarely, 2-3% on average per query.

Based on the observed values of $\rho$, and instances of over-ruling in adjudication we believe that annotators are able to consistently establish a relative similarity between papers and indicate strong agreement with an adjudicated set of ratings. Based on qualitative observations in the annotation process we noted the primary case of disagreement between annotators. Disagreements occurred most where a single facet was representative of multiple different finer-grained aspects. In these cases annotators initially focus on one of the aspects in making their annotations, when made aware of other aspects during adjudication, annotators readily accepted a different judgement. Appendix G presents an example. We believe this speaks to the effectiveness of an adjudication step. Finally, Figure 2, indicates the set of relevances for query abstracts annotated with two facets – we see that while some candidates are correlated in relevance others are only relevant to the query in one facet.

*Full-text vs abstract annotations*: To examine the effect of annotating the abstract of papers instead of full-text of papers we also conduct a small scale study to examine the extent of differences between the relevances produced by them. This is done by a single expert annotator annotating relevance based on abstract and full-text separately for 9 query abstracts (3 from each facet) and 5 candidate abstracts each (45 pairs). We refer readers to Appendix B for the details of the annotation setup. Agreement was measured between the relevances produced based on the abstract text and the full-text: Spearmans $\rho = 0.78$, Krippendorff's $\alpha = 0.77$, Cohens $\kappa = 0.63$, and % agreement of 73%. Full-text annotation was performed at the rate of about 5 minutes/pair, on the other hand abstract ratings took 15 seconds/pair. While our metrics indicates a less than perfect match between the abstract ratings vs full-text these metrics indicate strong agreement between the two. We believe this presents a reasonable trade-off between expense and completeness of annotation. Further, prior work on a similar scientific similarity annotation task noted low-agreements between expert annotators in annotation of full-texts [68, Sec 7], indicating that use of full-text does not necessarily translate into higher quality annotations.

**Comparison to Other Datasets:** To the best of our knowledge, Neves et al. [51] and Chan et al. [13] present the only other datasets of similar structure to the one we present:

**Neves 2019:** Neves et al. [51] annotate a set of 8 query papers with 70-90 candidates per query from biomedicine for similarity of the "papers research goal".
**SOLVENT:** Chan et al. [13, Sec 4.1] annotate a small scale dataset of 50 social computing papers for similarity along "purpose" or "mechanism" facets. Here, each of the 50 papers is annotated for relevance with respect to the other 49 papers in the set of 50 papers.

Both these datasets intend at presenting small scale evaluations in the context of other work. Our work, in contrast, presents a substantially larger resource created over multiple annotation passes, annotates similarity across multiple facets, and presents a reusable dataset. Several other datasets bear partial resemblances to the work we present here, we discuss these in Section 7.

**Annotation scalability:** We believe that a dataset like ours necessarily calls for expert annotation. Understanding the annotation guidelines alone required experience reading research papers. We believe these traits make it harder to scale the presented datasets using an un-trained crowd-sourcing based method. However, we believe certain aspects of our work can inform future work to scale datasets like ours. A dominant approach in building datasets involves annotation of instances by multiple annotators followed by a majority vote [9] or an average [3] to determine the "gold" label. Given the expense of creating repeated annotations when working with expert annotators we instead choose to use a adjudication step to create "gold" labels. We believe this helped us scale our approach considerably. If this step leveraged more experienced annotators to perform adjudication while lower experience annotators make initial judgements we believe it would be scalable beyond the approach presented here. This also follows prior work from Nallapati et al. [49], which leverages a hierarchy of annotator skills to scale a complex annotation task. Finally, we also note that large IR test collections call for competing systems to generate a pool of documents to judge [67]. We believe our dataset presents a robust evaluation set which may be leveraged for development of such methods.

## 5   Experimental Results

Here, we establish baseline performances from a handful of standard and state of the art methods.

**Baseline methods:**  The methods we choose to evaluate capture a range of granularities and nature of methods: term based methods, pre-trained model based sentence representations, and whole abstract representation models. Appendix C describes each method in detail.

Term-level baselines: `fabs_tfidf`, `fabs_bm25`, `fabs_cbow200`, and `fabs_tfidfcbow200` represent term-level baselines. These represent the document as sparse TF-IDF vectors, averaged bag of word representations, and weighted averaged bag of word representations respectively. Each of them represent the query document as the representation for the sentences corresponding to the query facet sentences in the query abstract. Candidates are represented by their whole abstract representations.

Sentence-level baselines: Here encode all query facet sentences and all candidate abstract sentences individually with a sentence encoder, and then use the maximum pairwise sentence cosine similarity between the query and candidate sentences to rank candidates. `SentBERT-PP`, `SentBERT-NLI`, `UnSimCSE`, `SuSimCSE` (unsupervised and supervised `SimCSE`) represent state of the art sentence encoder baseline models [56, 22]. Each of the models here represent models trained in a different manner or on different training sets.

Abstract-level baselines: SPECTER and SPECTER-ID represent whole abstract level representations [16]. Both of these approaches represents a multi-layer transformer based SCIBERT model fine-tuned on citation network data. SPECTER operates on titles and the whole abstract of the papers. Both queries and candidates are represented by their SPECTER embeddings. Note that SPECTER was trained on a corpus of randomly selected scientific documents. We also re-implement and train a version of SPECTER on about 660k computer science paper triples with identical hyper-parameters to SPECTER, we call this in-domain model SPECTER-ID.

In re-ranking we use the L2 distance between the query and candidate vectors unless noted otherwise. Note here that while the term-level baselines are more similar to the the task formulated in Definition 1, the sentence-level baselines solve the task in Definition 2. Further note that we make sure to include baseline methods from the above method types such that were not used for the constructions of pools. This is intended to evaluate the performance of methods which were not used for pool construction (§3), thereby investigating the ability of the dataset to be re-used for evaluating future methods. These are represented by `fabs_bm25`, `SentBERT` and `SimCSE` based methods, and the SPECTER-ID baseline.

**Re-ranking Results:** Table 3 denotes performance on the test set for each facet independently and aggregated in the *Aggregated* columns. In reporting results, we report R-Precision, Precision@20, recall@20, and NDCG@k. For NDCG@k, we follow Wang et al. [69], and choose $k = p * |\mathcal{C}|$ where

Table 3: Test set results for the set of baselines methods. Metrics (R-Precision, Precision and Recall at 20, NDCG$_{\%20}$) are computed based on a 2-fold cross-validation, represent averages over per-query metrics, and are reported as percentages. SPECTER-ID performance is reported over three training re-runs with underset standard-deviation, the remaining baselines are reported based on a single set of model parameters released by the respective authors.

| | background | | | | method | | | |
|---|---|---|---|---|---|---|---|---|
| | RP | P@20 | R@20 | NDCG$_{\%20}$ | RP | P@20 | R@20 | NDCG$_{\%20}$ |
| fabs_tfidf | 23.35 | 27.19 | 45.80 | 57.97 | 09.30 | 09.83 | 34.75 | 31.20 |
| fabs_bm25 | 20.12 | 27.81 | 49.85 | 59.39 | 09.37 | 11.63 | 38.29 | 34.59 |
| fabs_cbow200 | 19.61 | 15.94 | 27.97 | 36.56 | 08.65 | 08.33 | 15.69 | 21.14 |
| fabs_tfidfcbow200 | 15.92 | 16.87 | 27.76 | 40.51 | 07.99 | 06.01 | 17.71 | 21.70 |
| SentBERT-PP | 21.24 | 28.75 | 46.67 | 60.80 | 10.00 | 10.83 | 36.30 | 33.40 |
| SentBERT-NLI | 19.02 | 25.00 | 40.13 | 54.23 | 09.11 | 11.46 | 02.89 | 31.10 |
| UnSimCSE-BERT | 18.15 | 23.44 | 36.05 | 51.59 | 08.86 | 09.65 | 27.92 | 31.23 |
| SuSimCSE-BERT | 19.22 | 22.81 | 46.75 | 55.22 | 08.58 | 09.76 | 29.01 | 30.88 |
| SPECTER | **24.81** | **35.31** | **57.45** | 66.70 | **11.72** | 13.58 | 40.81 | 37.41 |
| SPECTER-ID | 24.55 ±1.3 | 34.17 ±0.5 | 53.26 ±0.3 | **69.22** ±1.71 | 10.53 ±0.3 | **16.22** ±1.21 | **44.59** ±3.6 | **42.76** ±0.78 |

| | result | | | | *Aggregated* | | | |
|---|---|---|---|---|---|---|---|---|
| | RP | P@20 | R@20 | NDCG$_{\%20}$ | RP | P@20 | R@20 | NDCG$_{\%20}$ |
| fabs_tfidf | 11.35 | 16.28 | 38.57 | 41.24 | 14.59 | 17.64 | 39.69 | 43.19 |
| fabs_bm25 | 11.31 | 20.00 | 40.40 | 45.07 | 13.50 | 19.69 | 42.73 | 46.06 |
| fabs_cbow200 | 11.16 | 10.42 | 23.44 | 30.93 | 13.08 | 11.47 | 22.23 | 29.36 |
| fabs_tfidfcbow200 | 10.43 | 10.69 | 24.39 | 32.79 | 11.38 | 11.09 | 23.13 | 31.42 |
| SentBERT-PP | 13.60 | 19.83 | 41.73 | 52.35 | 14.83 | 19.62 | 41.41 | 48.57 |
| SentBERT-NLI | 14.23 | 22.05 | 46.99 | 51.30 | 14.04 | 19.42 | 38.67 | 45.39 |
| UnSimCSE-BERT | 12.00 | 19.58 | 38.95 | 45.55 | 12.92 | 17.41 | 34.43 | 42.59 |
| SuSimCSE-BERT | 12.37 | 18.58 | 39.76 | 44.93 | 13.33 | 16.95 | 34.83 | 43.45 |
| SPECTER | 18.62 | 23.78 | 52.72 | 56.67 | 18.29 | 23.97 | 50.14 | 53.28 |
| SPECTER-ID | **20.09** ±0.92 | **27.36** ±0.45 | **58.74** ±3.04 | **60.40** ±1.31 | **18.32** ±0.79 | **25.74** ±0.22 | **52.12** ±1.54 | **57.22** ±0.70 |

$p \in (0,1)$. NDCG$_{\%20}$ therefore refers to NDCG computed at 20% of the pool size for a query. This is apt since our queries don't have identical pool sizes. Appendix F presents an extended result table.

*Overall results:* First, in line with the strong performance of pre-trained language model representations for a range of tasks, we note broadly the stronger performance of SPECTER-ID and SentBERT models compared to term and static embedding based baselines. However, note that the sentence level SentBERT models underperform models which incorporate the whole abstract context. Also note that training on in-domain data allows SPECTER-ID some gains over SPECTER. In examining facet dependent performance, we note the stronger performance of all the methods on the background facet, which is expected due to the stronger correlation between background sentences and the general topic of the paper. Next, we note the consistently poorer performance of all methods on the method facet, providing clear room for improvement. As might be noted from our relevance rating guidelines (§4), we rate "mechanistic" notions of similarity for the method facet. Given this relational nature of similarity, we expect methods relying on whole paragraph or term level representations to perform poorly on this facet. We note results midway between the other facets for result – this is due to some results being easy to be judged similar based on term overlaps while others require deeper a understanding of the query (See Appendix G). Finally, we make special note of the poor performance of recent state of the art models SimCSE, and SPECTER on overall performances, specially so in method and result facets – we believe this offers future work substantial room for improvement.

Since we annotate multiple queries per abstract we also present per-query results for the best performing baseline, SPECTER-ID, on this set of queries in Figure 3. We note here the difference in performance by facet for SPECTER-ID, an un-faceted model. Here, performing well on one facet does not always lead to strong performance on other facets indicating room for improvement with models which incorporate finer-grained conditional similarities into their rankings.

*Reusabilty:* Since we evaluate methods not used for pool construction (i.e fabs_bm25, SentBERT and SimCSE methods, and SPECTER-ID), we examine their performance. Here, both fabs_bm25 and SPECTER-ID outperform corresponding methods of their method-type used for pool construction (i.e. fabs_tfidf and SPECTER). Further SentBERT-PP, a method representing a different class of method than those used for pool construction also outperforms the kinds of methods used for pool construction, notably fabs_tfidf. We believe this indicates the lack of a serious bias of the dataset toward methods used for pool-construction, allowing re-use for evaluating future methods.

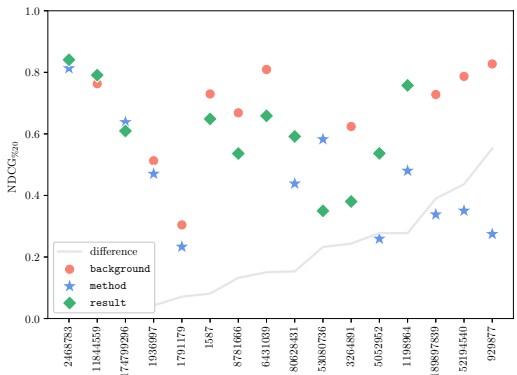

Figure 3: Per-query performance (NDCG$_{\%20}$) for SPECTER-ID on the set of papers which have been annotated with two facets (§3). Performances are means across 3 separate model training re-runs. The query papers are sorted by difference in performance on the facets.

# 6 Error Analysis

Based on a qualitative examination of per-query ranking performance of `abs_tfidf`, `SentBERT-PP` and SPECTER-ID we outline the factors which lead the baseline models to underperform. We believe the incorporation of modeling to handle these phenomena will lead to improved performance on the task. While we provide a summary of here, Appendix G provides more extensive examples.

*Salient Aspects:* One source of error is the inability of models to identify the most salient aspects for similarity – representations capturing "what the paper is about?", often expressed only in part of a larger set of facet sentences.

*Multiple Aspects:* Within a given facet, papers often expressed multiple finer grained aspects, models however often only retrieved based on a single aspect.

*Domain specific similarities*: A set of errors also arise from the inability of models to determine similarity between technical concepts. For example, an inability to judge "stacking", "ensemble strategy", and "bagging" as similar.

*Mechanistic similarities:* Nearly all methods perform poorly in the case of determining mechanistic similarity in `method` facets. This often relies on determining similarity across a sequence of actions. This problem also bears resemblance to the challenging setup of retrieval based on movie plots as in Arguello et al. [6].

*Context dependence of facets:* Faceted similarities as labelled here often also show context dependence on other facets, especially for `result` queries. Given that one major guideline for result similarity in our dataset is "the same finding or conclusion", being able to determine context similarity is important.

*Qualitative result statements:* `result` queries which summarize qualitative findings as, opposed to reports of performance on a dataset, often perform poorer, often requiring broader context and often lacking in term overlaps which may otherwise easily indicate relevance.

Therefore, the range of challenging phenomena captured in our dataset allow evaluation for novel modeling approaches to document similarity which in-turn translate into progress on a range of important problems. Appendix H indicates potential sources of training data which may be leveraged to train fine-grained document similarity models to overcome some of the above challenges.

# 7 Related Work

Work presented here ties to that of information retrieval and scientific literature search communities.

*Faceted Search*: IR has considered the task of faceted search, where facets have often been treated as fixed attributes of metadata [38], in line with a QBE setup our work provides a more semantic interpretation of a facet tied to the rhetorical structure of the document. Other similar work in IR comes from those aiming to diversify search results along the specific aspects/intents of an under-specified keyword query [4]; one might consider the "aspects" in this line of work to our "facets".

Others have also explored dynamic generation of facet like attributes for queries [39]. Partly similar to our task, Upadhyay et al. [65], allow specification of aspects along side ad-hoc queries.

*Query-by-Example*: A range of literature has also considered the QBE formulation applied to a variety of different kinds of data, from graphs, text, music and to archival image search [45, 2, 34]. Both Sarwar and Allan [59] and Taub Tabib et al. [62] frame retrieval from text corpora using event or syntactic representations as QBE. The closest work in QBE applied to research papers search comes from El-Arini and Guestrin [19], Jacovi et al. [31] which considers multiple document queries for research paper recommendation and Zhu and Wu [70], who additionally propose considering topic variety within multiple query documents. While El-Arini and Guestrin [19] evaluates their approach with a user study, Jacovi et al. [31] employ document key-phrases as a proxy for finer grained relevance.

*Literature search and recommendation:* Other related work to the one presented here comes from a range of work exploring literature search. Chakraborty et al. [12] trained faceted paper retrieval based on citation contexts in a paper. Jain et al. [32] train models to learn disentangled abstract representations trained on aspect-labelled data for biomedical randomized control trial papers. Chan et al. [13] also presents closely related work, where they explore the problem of recommending analogically similar scientific articles, and wherein they also include a small-scale evaluation dataset comparable to the one presented here. In similar vein Neves et al. [51], extensively evaluate a range of methods intended to extract rhetorical structure elements for faceted scientific paper search and evaluate it with a small scale dataset of biomedical publications labelled for fine grained facet similarity. Hope et al. [28] allows exploratory search using multi-facet characterizations of scientific articles for the COVID19 research literature. Faceted document similarity for articles has been explored most recently in by Ostendorff [53], Ostendorff et al. [54] and Kobayashi et al. [36]. Both Ostendorff et al. [54] and Kobayashi et al. [36] evaluate using the task of predicting facets of similarity for papers cited in a particular section of a research article. Kobayashi et al. [36] further evaluate on a context-dependent co-citation ranking task as well. Work presented in Ostendorff [53] and Kobayashi et al. [36] present tasks most similar to ours while lacking in manually annotated datasets.

A range of work also explores the problem of document representations for an unfaceted content based recommendations in scientific documents [16, 8] and are often evaluated using citation or paper recommendation tasks – Färber and Jatowt [21] provide an extensive survey of this literature.

*Other Datasets*: A handful of other work also bears resemblance to aspects of our dataset. Brown et al. [10], present a large expert annotated biomedical dataset intended to benchmark content based literature recommendation, however the dataset is annotated at the whole abstract level unlike faceted relevances as presented here. Datasets intended to match citation context sentences to the abstract sentences of the papers they cite share some similarity with the task of querying with sentences of a facet [14]. However they represent a somewhat simpler and different task, requiring to match sentences from full-text to the citance. In similar vein, our task setup also resembles that of claim verification as proposed in Wadden et al. [68]. This setup however while serving the different goal of claim verifications also deals with atomic facts as opposed to more complex context dependent scientific paper facets as in our setup. Finally, datasets intended for citation intent prediction [33, 50, 63] mainly focus on a classification task involving predicting citation-intent given pairs of papers as opposed to retrieving papers while conditioning on a paper and a facet.

## 8   Conclusions

In this work we formalize the task of faceted Query by Example in the context of scientific literature search and highlight several important related problems our dataset can facilitate progress on. Given the problems of scientific search, the inability to articulate keyword queries, context dependence, and desire for exploratory search, we believe the faceted QBE formulation provides meaningful benefit. We provide a expert annotated test collection for the evaluation of the faceted QBE task. While prior work on the problem has often relied upon small evaluation sets, silver evaluations based on citations or keyword based relevance, or expensive human evaluation we believe our dataset provides a pragmatic alternative which will facilitate comparison and development of models. Finally we evaluate performance with a host of strong baseline approaches and highlight the challenging aspects of the dataset and the faceted QBE problem in general, and note that the dataset offers significant room for improvement.

# 9 Acknowledgements

We are grateful to the members of IESL at UMass Amherst and the Olivetti Group at MIT for helpful discussions in formulating this project. We also appreciate the contributions of our annotators in creating the annotation guidelines and the dataset. We also thank anonymous reviewers for their suggestions for making our work stronger. This work was supported in part by the National Science Foundation under Grant No. DMR-1922090, the Chan Zuckerberg Initiative under the project Scientific Knowledge Base Construction, the International Business Machines Corporation Cognitive Horizons Network under the project Knowledge Extraction, Representation, and Reasoning, and the Center for Intelligent Information Retrieval. Any opinions, findings and conclusions or recommendations expressed in this material are those of the authors and do not necessarily reflect those of the sponsors.

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
