# OpenReview forum: "CSFCube - A Test Collection of Computer Science Research Articles for Faceted Query by Example"
_NeurIPS.cc/2021/Track/Datasets_and_Benchmarks/Round2 — NeurIPS 2021 Datasets and Benchmarks Track (Round 2)_

### Official Review · Reviewer_aBb2 · 2021-09-19
**A useful dataset for a challenging query-by-example retrieval task**

**Rating:** 5
**Confidence:** 4

**Strengths:**

- A successful non-trivial annotation study has been reported
- Good license choice

**Weaknesses:**

- Clarity issues
- Dataset comparison issues
- Documentation issues
- Discussion on scalability is missing

**Additional Feedback:**

This paper is a resubmission from the CIKM 2021 Resources Track.

**Clarity:**

It was unclear how did the authors compute the NDCG values. The higher the number of elements, the higher its value is. Have the authors used NDCG@k, and if so, the authors should discuss the choice of the `k` parameter. I recommend citing the discussion in the Wang et al. (2013) paper: https://proceedings.mlr.press/v30/Wang13.html.

As the authors have performed manual annotation of the presented dataset, I am interested in reading the discussion on the scalability of the proposed annotation approach. How easily can one create a similar dataset using machine translation or crowdsourcing?

It was also unclear which data items caused the most confusion by the annotators to avoid such issues in the future.

Was it possible to use nearest neighbor search as a baseline?

**Correctness:**

The authors have reported Spearman correlation coefficients as agreements, which is rarely used for this. Instead, I recommend reporting Krippendorff's alpha to evaluate the inter-rater agreement.

**Documentation:**

The authors offer neither evaluation code nor documentation for using their dataset, which is crucial for the datasets and benchmarks paper. From the repository, one should clearly understand how simple it is to evaluate their method on the benchmark proposed by the authors.

**Ethics:**

I see no specific ethical concerns in the reported study.

**Relation To Prior Work:**

In Section 7, the authors mention several related datasets but offer neither qualitative nor quantitative comparison to their contribution. I recommend providing such a comparison.

**Summary And Contributions:**

This paper proposes CSFCube, a novel dataset for evaluating query-by-example retrieval methods in the computer science literature domain. Although I appreciate the authors' efforts in producing the dataset and evaluating the methods, the present version of the paper lacks description clarity. Also, the presented repository does not offer convenient means for method benchmarking.

---

> ### Author Response · Authors · 2021-09-27
> **Response to R4.**
>
> **Inter-annotator agreement (re: Correctness):** We will add Krippendorff's alpha to the paper. These values before and after adjudication are indicated below (computed using: [kripp.alpha](https://rdrr.io/cran/irr/man/kripp.alpha.html) with ordinal distance):
> ```
> background: pre 0.4288, post 0.7201;
> method: pre 0.2557, post 0.6107;
> result: pre 0.3518 post 0.6687.
> ```
> Post adjudication metrics are computed as per Lines 208-210. We would also like to point out that while our use of Spearmans rank correlation is uncommon we believe it is a sound choice. Since we present a ranking task, we are most interested in the rankings over candidate documents produced by annotators; a strong correlation in the rankings of documents indicates similar understandings of relative similarity between annotators. Spearmans rank correlation captures this intuition.
>
> **NDCG computation (re: Clarity 1):** Thank you for this note. We will update the paper with NDCG computed for lower k. Following the paper you linked, [Wang et al. 2013, Sec 3.3](https://proceedings.mlr.press/v30/Wang13.pdf), we choose k to be 20% of the pool size i.e. $k=c*n$ where $c \in (0,1)$ and n is pool size. In re-computation we noted that trends across models remained the same, the magnitude of NDCG was lower, and the differences between models was also larger. Current NDCG in the paper was computed over the whole candidate pool as per $DCG_k=rel_1+\sum_{i=2}^{k}\frac{rel_i}{log_2(i)}$ with NDCG computed as $\frac{DCG_k}{IDCG_k}$ where IDCG is the DCG for an ideal ranking.
>
> **Scalability of approach (re: Clarity 2):**
> *Using crowdsourcing:* Given the expertise necessary to understand research articles we dont believe a non-expert crowd can create a dataset of this nature. But we believe with an expert-crowd (computer science graduate and above), some training, and sufficient resources (time and expense), it would be possible to scale the approach here to create a dataset atleast an order of magnitude larger. We believe an adjudication stage would be helpful to scale the dataset creation, where lower-experience raters can create initial sets of ratings with a more experienced rater making adjudication decisions. Also note that a large dataset would require organization of a shared task with competing systems, we believe our dataset in its current form would benefit the development of these competing systems based on which future larger datasets may be built.
> *Using machine translation:* It isn't clear how one might apply machine translation in building a dataset like this, could you please elaborate?
>
> **Data-items causing confusion (re: Clarity 3):** We describe cases of frequent disagreement in lines 216-220, and present an example in lines 762-768. Initial disagreements were observed most often when annotators focused on different finer-grained aspects of a target facet. However, in these cases, annotators readily accept alternative similarity judgements during adjudication. We believe an adjudication step helps overcome these disagreements.
>
> **Nearest neighbour search baseline (re: Clarity 4):** All of our baselines are based on a document representation (e.g. sparse, word-embedding based, state of the art representations with SPECTER, etc) combined with exact nearest neighbor search. Please let us know if you had a specific other baseline in mind.
>
> **Prior datasets (re: Relation to prior work):** The paper presents qualitative and quantitative comparisons between the proposed dataset and the closest prior work on Lines 221-231 (Section 4). Datasets mentioned in Section 7 (Related work) are significantly different but were highlighted for partial similarities to our work. Please let us know if you would like us to add discussion for these datasets in addition to those in Section 4, we will update the paper with this.
>
> **Usage instructions (re: Documentation):** We will add instructions and code examples for using the proposed dataset.

---

### Official Review · Reviewer_P6Gt · 2021-09-20
**The dataset will help researchers working on scientific research analysis by providing ground-truth similarity for research papers. The paper is well written and their methodology has the required rigour.**

**Rating:** 7
**Confidence:** 5

**Strengths:**

Scientific research analysis has become a popular research topic, and this dataset will have an impact in that field. There is a gap in the community in obtaining ground-truth data. The dataset has been made very accessible on github.

**Weaknesses:**

The paper is quite strong overall. However, the overall result that baselines perform poorly on the produced dataset, and so, it provides opportunity for improvement to the community, is not fully established in my opinion. This is because there is sufficient disagreement among human annotators too, that needed adjudication stage to rectify. So, it is not clear whether there is a real ground truth at all, or there is sufficient scope for multiple views in the similarity measurement process. Despite this, I think the dataset is a well-needed attempt, and its utility will need to be put to the test of time.

**Additional Feedback:**

1. Def can be put in its full-form: Definition.
2. In line 145: "The remaining 18..."; but 50-16 is not 18!


**Clarity:**

Clear. I feel Appendix E or a part of it, is important enough to become part of the main paper, if space can be accommodated.

**Correctness:**

The dataset is constructed in a sound way. After getting candidate similar pairs based on automated means, the pairs have been filtered and annotated manually, with sufficient redundancy and supervision for robustness.

**Documentation:**

Sufficient.

**Ethics:**

No concerns. No privacy issues are involved.

**Relation To Prior Work:**

Differences from previous contributions are discussed.

**Summary And Contributions:**

The dataset lists similar papers for a given query paper, where similarity is seen based on one of three facets (objective, method, result). ground-truth similarity for research papers. After getting candidate similar pairs based on automated means, the pairs have been filtered and annotated manually, with sufficient redundancy and supervision for robustness. The paper is well written.

---

> ### Author Response · Authors · 2021-09-27
> **Response to R3.**
>
> **Existence of ground truth (re: weakness 1):** You are correct in noting that multiple views on similarity might exist. While this is true, we noted in the adjudication process that multiple views on similarity primarily existed due to annotators focusing on different finer-grained aspects of a target facet. Lines 762-768 presents an example. In these cases annotators readily accepted alternative similarity judgements indicating against the serious lack of a ground truth. Further, note that the situation where the adjudication overturned both annotator judgements was very rare (2-3% candidates). We believe that this indicates the existence of a reasonable ground truth within the scope of our defined guidelines.
>
> As further analysis, we also estimated “human performance” on the dataset by treating our initial raters judgements as predictions and the adjudicated ratings as a ground truth (since two raters independently judge candidates, we averaged raters evaluation metric values per query). These metrics are shown below:
> ```
> background: R-Precision; Precision@20; Recall@20; NDCG
> 0.3711, 0.4422, 0.7230, 0.9174
> method: R-Precision; Precision@20; Recall@20; NDCG
> 0.2923, 0.2530, 0.7697, 0.8305
> result: R-Precision; Precision@20; Recall@20; NDCG
> 0.2565, 0.3677, 0.7916, 0.8685
> all: R-Precision; Precision@20; Recall@20; NDCG
> 0.3050, 0.3523, 0.7615, 0.8714
> ```
> Note that these are significantly higher than the best baselines. Though these are also likely to be somewhat biased toward human ratings since the “gold” ratings are adjudicated based on initial human ratings. Still, we believe this indicates that even with an imperfect/variable ability to judge similarity, human raters perform much better than the best baselines indicating that there is progress to be made. Finally, we point out that while raters may not agree in absolute terms on a similarity level we are concerned most with ability to judge relative similarity between papers, we believe our Spearman rank correlation metrics indicate that raters relative similarity judgements are correlated.
>
> **Clarification for line 145:** We annotated 16 query abstracts against candidates for 2 different facets each, e.g. the query abstract with paper-id 11844559 was annotated for background and result facets. So the 18 comes from 50 - (16 x 2).

---

### Official Review · Reviewer_yLXs · 2021-09-20
**A useful test collection for query-by-example task with concerns about its reusability and judgment accuracy**

**Rating:** 8
**Confidence:** 5

**Strengths:**

The test collection will be useful for information retrieval (IR) research.
They share their data and annotation guideline.
They claim that they will release the code of baseline methods they implemented after acceptance of the paper.
Their annotation guideline is comprehensive.
They conduct a qualitative analysis for errors of baselines.
The paper is well-written.


**Weaknesses:**

•	The authors provide judgments for 6244 query-document pairs. Considering the time-consuming nature of judging process, the size of the collection is acceptable. However, there are many unjudged documents and many systems which have not participated the pool, might return unjudged documents. Normally, these documents are assumed to be non-relevant. Therefore, I would love to see reusability analysis of the test collection. However, I accept that it is not easy when there is no shared-task to construct the pool. Therefore, at least a discussion about reusability would be good.

•	The authors judge query-document pairs based on abstracts of papers. This sounds reasonable, considering the cost of judging based on all papers. However, this judging might also cause incorrect judgment because the information in abstracts might be limited. Therefore, I would love to see whether judgments based on abstracts are in line with judgments based on full papers. In order to do that they could just sample some query-document papers and do their analysis.

•	The authors report Spearman rank-correlation and agreement ratio to show the agreement on relevance judgments, instead of popular Cohen’s Kappa score. The authors justify this because Cohen’s Kappa does not consider ordinal values but Spearman rank-correlation does. I agree on this and it makes sense. However, Cohen’s Kappa scores are easily interpretable. It is not clear to me whether spearman rank-correlation scores they report are good enough. Instead of reporting just agreement ratio, they could report Cohen’s Kappa.

•	As baseline systems, they use TF-IDF as one of their approaches. I would prefer BM25 since it usually outperforms standard TF-IDF.


•	Their pooling method is not the standard pooling method such that they get 33 or 13 results from each of the retrieval results and at each query the number of papers drawed from a system can change (either 13 or 33). It is not clear to me why they did so. They could use a fixed pool depth such that the total number of judged documents are 100 (or close to 100). This way the evaluation of participating methods would be fair. In this way, it might not be fair.


**Additional Feedback:**

In the revision process, the authors addressed all points I raised. I appreciate their effort for these. I changed my rating from 6 to 8.

**Clarity:**

The paper is well-written. There are only a few typos.

- In section 2: “….pre-defined facets [26, p14], we expect QBE” p14?
- “…. this happen a very rarely, …”
- In Section 7, there are several singular-plural mismatches.
- “from full-text to the citance.”  Citance?
- Colors in Figure 2 are not interpretable.


**Correctness:**

There are no major mistakes in terms of correctness. However, there are some issues as mentioned above in Weaknesses section. These are fairness among participants, judging based on abstracts without analyzing its impact and not reporting Cohen’s Kappa score.

**Documentation:**

There is enough detail on data collection, etc. The dataset is available. The implementation of baselines is not available yet but the authors say they will release it in the camera-ready version.

**Ethics:**

There is no ethical issue.

**Relation To Prior Work:**

The authors provide a detailed discussion about related work. It seems that their work makes a solid contribution compared to prior work.

**Summary And Contributions:**

Authors construct a new test collection for Query By Example (QBE) search. They use abstracts of 50 papers as queries and judge them based on different facets including method, background, and result. In order to select the papers to be judged, they do pooling using methods they developed. In total, they judge 6244 query-document pairs. They also provide performance of 9 different methods.  They release their data and annotation guidelines

---

> ### Author Response · Authors · 2021-09-27
> **Response to R2.**
>
> **Clarification on pooling (re: Weakness 5):** We apologize for this misunderstanding, the paper is not worded correctly. We will change the writing to reflect this. Our pooling method did NOT use the top 33 or 13 results from a system causing the number of results from a system to change per query. We used the same pool depth for all the methods for a given query. Our dataset contains 8 queries with a pool size of ~250 and 42 queries with pool size of ~100. For the 8 queries with a pool of size ~250 we used a pool depth of 33 and for the 42 queries of pool size ~100 we used a pooling depth of 13.
>
> **BM25 instead of TF-IDF (re: Weakness 4):** We will update the paper with BM25 baseline performance. BM25 does indeed to a few points better than TF-IDF but the result trends and conclusions remain the same.
>
> **Cohens Kappa vs Spearman (re: Weakness 3):** Given that the spearman rank correlations are bounded between 1 and -1, we interpreted the reported rank correlations between annotators, especially post adjudication (0.77, 0.84, 0.83) as a sign that the rankings of candidates produced by annotators was highly correlated (with a “gold” adjudicated set) indicating significant agreement of the relative similarity of documents.
> Cohens Kappa scores per facet before and after adjudications are as follows:
> ```
> background: pre 0.2814, post 0.6214;
> method: pre 0.2029, post 0.5371;
> result: pre 0.2632 post 0.5903.
> ```
> Cohens Kappa scores will be added to the paper (also see Krippendorff's alpha in Response to R4). Post adjudication metrics are computed as per Lines 208-210. Note that agreement post adjudication (the ratings used for evaluation) are high. Adjudication accounts for mis-calibration in the interpretation of scores by initial annotators which is one of the factors hurting a pre-adjudication kappa. Another factor of disagreement which is easily fixed in adjudication, as described in lines 216-220 (lines 762-768 present an example), are where annotators readily accept alternative similarity judgements because of spans of text in the query and candidate which they failed to focus on in determining their initial judgements.
>
> **Full text vs abstract (re: Weakness 2):** Thank you for this suggestion. An analysis of a handful candidates and queries full text vs abstract based ratings will be added to the paper. However, we would also like to note that in selecting query abstracts we manually make sure that all query abstracts contain information in sufficient detail to allow reasonable judgements of similarity. Similarly, candidate abstracts were chosen to be of sufficient length as well. Further, we emphasize that the expense of annotating fine-grained similarity at the level of full texts would be high enough to make it infeasible, necessitating the approach followed here and in numerous prior work (lines 77-81).
>
> **Re-usability (re: Weakness 1):** A discussion on re-usability will be added. Note that in the current set of results we make sure to include methods which were not used in the construction of the pools. These are represented by SentBert-\*, SimCSE-\*, and SPECTER-ID where the first two methods represent significantly different types of methods than the ones used for pooling. Yet these methods are able to outperform some methods used for pooling. This indicates the lack of a strong bias toward pool methods and that non-pool methods can be evaluated using our dataset.
>
> **Responses to clarity issues:** We will clarify/fix these in the paper.
> - In section 2: “….pre-defined facets [26, p14], we expect QBE” p14? - p14 specifically references page 14 of the cited paper.
> - “from full-text to the citance.” Citance? - Citance refers to the sentence making a citation, as per usage by [Nakov et al.](https://biotext.berkeley.edu/papers/citances-nlpbio04.pdf), [Wadden et al.](https://arxiv.org/abs/2004.14974) and others.

---

> ### Author Response · Authors · 2021-10-04
> **Thank you!**
>
> We appreciate you updating your score and flagging the gaps in our initial submission. We believe it made our work much stronger! Thank you!

---

### Official Review · Reviewer_aye2 · 2021-09-23
**Carefully constructed QBD dataset with unclear impact**

**Rating:** 4
**Confidence:** 4
**Correctness:** The claims in the paper are correct t…

**Strengths:**


S1: The test collection is carefully constructed using standard IR procedures.

S2: The error analysis section of the paper is worthy of appreciation. It makes the complexity of the task much clearer.

S3: The writing is clear and is understandable.

**Weaknesses:**

W1: The baselines considered are somehow simplistic and are not always based on established IR QBD baselines. This somehow also shows the lack of work in this area. This is also a concern in general for the benchmark. It seems like this is a niche IR task and might not garner enough interest from the ML community or the IR community at large. The authors claim the potential importance of the task but do not come up with effective QBD baselines in the literature "modified" for aspect modeling.

W2: The aim of the test collection/benchmark is to promote interest in ML-based techniques and more specifically representation learning approaches to make progress in QBD tasks. However, it is not sufficiently clear that the scale of the dataset or the query workload is reasonably large to train supervised models. For a first dataset, a smaller dataset means that one has to heavily rely on task-specific modelling rather than transfer learning. The authors should reflect on this limitation of their collection.


W3: Simple baselines seem to be already quite close to the abstract-based representation approaches. On one hand, this means that simple embedding-based or transfer-learning-based approaches are not successful. On the other hand, it is also not clear how existing QBD approaches (modified for aspect-based querying) would compare against simple term matching baselines.

**Additional Feedback:**

no additional concerns

**Clarity:**

The paper is well written and clear for most parts. The evaluation metrics could have been better clarified. Specifically, how were the ranking metrics calculated ? I failed to understand how the relevance assessment would be carried out between a selected passages and the ground truth passage, and how it could be used in the NDCG computation.

**Documentation:**

The documentation is adequate, and the contents are accessible.

**Ethics:**

No ethical concerns.

**Relation To Prior Work:**

Most of the related work is well covered.

**Summary And Contributions:**

CSFCube


This paper builds a test collection CSFCube for the "Query by Example" task in IR. QBE tasks take a document as a query and long documents tend to have multiple facets (much more than ambiguous queries). The authors, restrict the QBE task to a faceted QBE task in which users can specify a fine-grained aspect in addition to the query document. The dataset is curated with a scientific literature search use case in mind where the facets are rhetorical content structure units like objective, results, approach, etc,.
The test collection consists of a smaller set of 50 documents using best practices from TREC i.e. using pooling with high-quality graded relevance scores. The authors also present experiments using term-based methods, pre-trained model-based sentence representations, and abstract-representation models. The abstract representation-based methods exhibit the most superior and the other embedding-based baseline suffer showing off-the-shelf models and approaches are not enough to achieve reasonable performance showcasing the difficulty of the task.

---

> ### Author Response · Authors · 2021-09-27
> **Response to R1.**
>
> **Impact of the task:** While we do not present aspect dependent models in this dataset track paper, we believe the dataset to be a meaningful resource to a number of communities. For the areas of ML and IR while we dont expect sole focus on our dataset we believe it presents a robustly annotated challenge evaluation set which may be included in broader benchmarks. Further, as we highlight in Lines 721-743, we expect a number of important problems outside those of ML and IR to benefit from this work: expertise search in the context of peer review which relies upon fine-grained similarities of papers, exploratory search for scientific literature, and problems in science studies. Finally, we note that SOTA representation learning methods such as SPECTER are currently used in paper-reviewer matching on Openreview ([link1](https://www.science.org/news/2021/04/ai-conferences-use-ai-assign-papers-reviewers), [link2](https://github.com/openreview/openreview-expertise)), and paper-recommendations for scholarly search engines such as Semantic Scholar ([link3](https://twitter.com/semanticscholar/status/1267867735318355968)). Progress in fine-grained document similarity, of the kind which can be measured on our dataset, for methods such as SPECTER have the potential to transfer to immediately relevant and important services.
>
> **Existing query by document (QBD) baselines (re: W3):** You correctly note that the intention for our work is to encourage work in representation learning approaches to make progress in QBD tasks. We would like to note that the SPECTER baselines (SPECTER and SPECTER-ID), based on a whole abstract representation, present a SOTA document representation for QBD tasks (as opposed to a faceted/aspect based QBD task). These methods perform significantly better than term based approaches (~6-10 points on NDCG and larger margins on other metrics across all facets) indicating that existing SOTA QBD approaches outperform term based baselines and yet leave substantial room for future work.
>
> **Modifications to QBD models (re: W1 & W3):** Follow up work since this submission has indicated that it is possible to outperform baselines but requires significant modeling and training data contributions. These approaches are under review in other modeling focussed venues. While our paper does not report this, at the time of this submission, we found that simple modifications to a SOTA QBD approach to capture aspects did not outperform baselines, but performed similar or worse than SPECTER-ID in the paper. Broadly, these approaches used SPECTER to obtain facet representations instead of abstract representations and were fine-tuned on citation network data to make aspect level matches across documents. We would be happy to add failed approaches based on extensions to SPECTER to this paper but we refrained from doing so given this venue's dataset development focus.
>
> **Clarification on computation of metrics (re: Clarity):** The relevance judgements were made by showing annotators a pair of documents which consists of query and candidate abstracts, and a target facet (Fig 1 of the annotator guidelines shows the interface: [link](https://github.com/iesl/CSFCube/blob/master/ann_guidelines.pdf)). Annotators rate the relevance between the sentences indicative of the target facet while considering the whole abstract context on a graded scale (Sec 4 of the paper describes relevance levels). At the time of evaluation, models rank candidate documents given the query abstract and the target facet. NDCG is computed using the ranked documents and the graded relevance judgements as per $DCG_k=rel_1+\sum_{i=2}^{k}\frac{rel_i}{log_2(i)}$ with NDCG computed as $\frac{DCG_k}{IDCG_k}$ where IDCG is the DCG for an ideal ranking. An evaluation script will be added to the dataset release.
>
> **Training on our dataset (re: W2):** We clarify that we do not intend for the dataset to be used for training. One of the primary motivations of the dataset is to facilitate a robust evaluation of faceted query by example and fine-grained similarity in the context of scientific papers. Lines 51-62 elaborate on the need for such an evaluation dataset. Further, we note that our best performing baseline (SPECTER-ID) is infact trained on weakly supervised data from the citation network, given current modeling trends of self-supervised learning we believe meaningful progress is possible without a labelled training set, Lines 813-827 suggests potential sources of training data one might leverage for improvements on our dataset.

---

### Author Response · Authors · 2021-09-27
**We appreciate the reviews!**

Dear reviewers, thank you so much for your time, the detailed notes, the suggestions for improvement, and the pointers to relevant work. We have posted an initial set of responses to all of your comments and are currently working on incorporating all the suggested changes into the paper. The paper will be updated shortly. The following is a summary of the additions we will be making based on the reviews:
1. Small scale analysis of abstract vs full-text relevance ratings. (as per R2)
2. Reporting Cohen’s Kappa and Krippendorff's alpha in addition to Spearman rank correlations. (as per R2 and R4)
3. BM25 baseline in addition to TF-IDF. (as per R2)
4. Clarifications on NDCG computation and reporting NDCG at lower values of K. (as per R1 and R4)
5. Instructions and code examples for using the dataset. (as per R4)
6. Important clarifications and discussion additions to the paper based on the reviews. (R1-R4)

Please let us know if any further additions/clarifications will be necessary. We hope our responses and additions convince you to update your scores.

---

### Author Response · Authors · 2021-09-29
**Paper has been updated with all reviewer requests.**

We have now completed updating the paper with all of the suggestions from reviewers, with clarifications made in individual responses. We deeply appreciate all the feedback and believe that it has strengthened our paper and reinforced our contributions. Here is a summary of changes:
1.  Line 232-245 and 744-755: A small scale analysis of full-text vs abstract based annotations was added. Agreements between abstract vs full-text ratings indicated strong agreement between the two ratings. (as per R2)
2. Table 2 now reports Cohen’s Kappa and Krippendorff's alpha in addition to Spearman rank correlations. Lines 210-221 contextualize these metrics. Agreements of annotators with adjudicated annotations indicates high agreement. (as per R2 and R4)
3. Table 3 now reports results for BM25 as a baseline. While out-performing TF-IDF, the results and conclusions remain unchanged. (as per R2)
4. Lines 303-308 and Table 3: Add clarifications regarding NDCG computation. Further our table now reports NDCG at lower values of K. Lower-values of K reduce the NDCG values while broadening gaps between baseline methods even further, and reinforce conclusions in the paper about potential for future work. (as per R1 and R4)
5. Lines 330-337: Discuss re-usability of the proposed dataset. (as per R2)
6. Lines 257-270: Discuss the scalability of the annotations. (as per R4)
7. Lines 155-173: Clarifies the method of candidate pooling for similarity annotation, importantly, this corrects from an incorrectly phrased  sentence which called into question the correctness of our pooling method. (as per R2)
8. The dataset repository will be updated with usage instructions shortly. (as per R4)

We hope you find the changes compelling. Please let us know any additional thoughts.

---

### Decision · Program_Chairs · 2021-10-09

**Decision:**

Accept

**Comment:**

This paper presents a dataset of annotated abstracts for faceted query by example (QBE) task. The new task is very interesting, the dataset, albeit not too large, is well annotated and clearly described. This paper will be an important contribution to the field of IR and will serve as a starting point into finer-grain and thus more useful QBE. While some reviewers had some concerns, the author response, as well as the revised paper, addressed most of the concerns.